# Quantifying the Aerodynamic Power Required for Flight and Testing for Adaptive Wind Drift in Passion-Vine Butterflies *Heliconius sara* (Lepidoptera: Nymphalidae)

**DOI:** 10.3390/insects14020112

**Published:** 2023-01-21

**Authors:** Robert B. Srygley, Robert Dudley, Edgar J. Hernandez, Franz Kainz, Andre J. Riveros, Charlie P. Ellington

**Affiliations:** 1Smithsonian Tropical Research Institute, Balboa 0843-03092, Panama; 2Pest Management Research Unit, Agricultural Research Service, 1500 N. Central Ave., Sidney, MT 59270, USA; 3Department of Integrative Biology, University of California-Berkeley, Berkeley, CA 94720, USA; 4Department of Biomedical Informatics, University of Utah, 421 Wakara Way, Salt Lake City, UT 84108, USA; 5Maximilian-Kolbe Straße 26, 61440 Oberursel, Germany; 6Departamento de Biología, Facultad de Ciencias Naturales, Universidad del Rosario, Bogota 111221, Colombia; 7Department of Zoology, University of Cambridge, Cambridge CB2 3EJ, UK

**Keywords:** biomechanics, behavioral ecology, free flight, rainforest, locomotion, kinematics, migration

## Abstract

**Simple Summary:**

Theory predicts that flying animals maximizing their migratory distance should adjust their airspeeds for headwinds and tailwinds. This energy-conserving adjustment in airspeed is based on the theory that as an insect increases its airspeed, the power required to fly increases rapidly above a minimum power velocity. Here we sought to quantify the aerodynamic power required to fly at different speeds in a migratory butterfly *Heliconius sara*, and predict the adjustment in airspeed necessary to minimize energy consumption in headwinds or tailwinds. We filmed butterflies as they migrated naturally using two high-speed video cameras. Simultaneously we measured the butterflies’ airspeeds while they flew across the Panama Canal, and we captured the same butterflies to measure weight distribution and the shape of the body and wings. We found that the power required to fly increased non-linearly from a minimum power velocity of 0.9 m/s. Faced with a headwind or tailwind, the butterflies increased or decreased their airspeed in accordance with that predicted. However, the null hypothesis of no compensation could not be rejected. This approach can be applied to predict insect movement between agricultural patches and the maximization of fuel efficiency when migrating overseas.

**Abstract:**

Although theoretical work on optimal migration has been largely restricted to birds, relevant free-flight data are now becoming available for migratory insects. Here we report, for the first time in passion-vine butterflies, that *Heliconius sara* migrates directionally. To test optimal migration models for insects, we quantified the aerodynamic power curve for free-flying *H. sara* as they migrated across the Panama Canal. Using synchronized stereo-images from high-speed video cameras, we reconstructed three-dimensional flight kinematics of *H. sara* migrating naturally across the Panama Canal. We also reconstructed flight kinematics from a single-camera view of butterflies flying through a flight tunnel. We calculated the power requirements for flight for *H. sara* over a range of flight velocities. The relationship between aerodynamic power and velocity was “J”-shaped across the measured velocities with a minimum power velocity of 0.9 m/s and a maximum range velocity of 2.25 m/s. Migrating *H. sara* did not compensate for crosswind drift. Changes in airspeed with tailwind drift were consistent with the null hypothesis that *H. sara* did not compensate for tailwind drift, but they were also not significantly different from those predicted to maximize the migratory range of the insects.

## 1. Introduction

In long distance migration, energy is at a premium and natural selection for conserving energy is potentially strong. Animals may adopt behaviors while in transit to reduce the energetic cost of flight between a locale of origin and their destination [1]. However, minimizing energetic costs is only one means by which animals may migrate optimally. Other strategies include minimizing time or minimizing risk of mortality during flight. Theoretical work on optimal migration has been largely restricted to birds [2]. More recently, optimal migration models have been adapted to flying insects [3,4,5,6,7].

Because many migrating butterfly species fly only 1–2 m above the Earth’s surface, an individual’s flight components, such as airspeed, heading, local wind speed, and wind direction, can be measured simultaneously [8]. With these, data optimal migration models can be tested directly. For example, male and female cloudless sulphur butterflies, *Phoebis sennae*, may have different optimal migration strategies. One theoretical prediction is that in order to maximize the distance flown on a given energy reserve, migrants should reduce their airspeed in a tailwind and increase it in a headwind. As predicted, female *Phoebis sennae* butterflies migrating over the Caribbean Sea slowed their airspeed in a tailwind and increased it in a headwind, whereas males did not adjust their airspeed for tailwind drift [3]. Females may minimize energy consumption to conserve lipids for eggs, whereas males may minimize the time to the destination site in order to maximize the opportunities to mate with newly arrived or newly emerged females [3].

Thus far, we have assumed that butterflies are operating with a “U”-shaped power curve in which hovering and flying fast are more costly than flying at moderate speeds. This assumption is qualitatively correct and is based on the mechanical power curve for migrating *Urania fulgens* moths [9]. However, quantitative analysis of changes in kinematics and aerodynamics with flight speed are lacking for any butterfly species in free, natural flight [10,11].

The passion-vine butterflies in the genus *Heliconius* are a model for investigating the role of flight in mimicry, courtship, and reproductive isolation [12,13,14]. For this paper, our objective was to measure the flight kinematics over a range of velocities for the passion-vine butterfly *Heliconius sara*, and estimate the aerodynamic power requirements for flight to test quantitative predictions of adaptive wind drift.

## 2. Materials and Methods

### 2.1. Study Organism

*Heliconius sara* (Figure 1) has a range from Mexico to Brazil. The species is distasteful and flies slowly, because it is protected by cyanogenic compounds that are manufactured *de novo* and others that are sequestered as larvae from their hostplant *Passiflora auriculata* [15]. The butterflies annually fly across central Panama near the beginning of the wet season. The track directions of *H. sara* flying across the Panama Canal near Barro Colorado Island (latitude 9° 09′ N; longitude 79° 51′ W) between 1990–2004 were distributed axially with mean vectors of 48° and 228° (Rayleigh’s test, r = 0.42, n = 21, *p* < 0.025). Near Barro Colorado Island where this study took place, the Panama Canal cuts through a large artificial reservoir called Lake Gatun. As a result, ‘lake’ and ‘canal’ are used interchangeably in this paper.

### 2.2. Filming of Natural Migratory Flight

A pilot, a camera-person, and camera-assistant paced individual migrating butterflies in a small boat (4.3 m aluminum dinghy with a 40 hp outboard motor) piloted so that the insect was positioned approximately 0.5 m off the port bow. The camera-person filmed the motion of the butterfly’s body and wings at 250 frames per second (fps) with two synchronized high-speed video cameras (Redlake Motionmeter 500), each fitted with a Cosmicar 6 mm television lens. The cameras were stereo-mounted 91 cm apart on a hand-held, steel slide bar (Berezin Stereophotography, 96.5 cm long × 7.6 cm wide) so that the lenses were orthogonal. During filming, the assistant measured the forward airspeed of the insect with a hand-held hotwire anemometer (Omega HHF52) while the pilot measured ground speed and direction with a marine differential GPS receiver (Garmin 152). Following capture of the butterfly with a hand net, the assistant measured ambient wind speed with the anemometer and direction with a wind vane and compass. Video sequences were stored on Sony digital videotapes and later transferred to a computer.

### 2.3. Morphology 

Each butterfly that was captured was identified, sexed, and frozen within two hours of capture. Morphological features relevant to flight (body length, body mass, center of body mass, and radial moment of inertia of the body, wing length, wing area, mean chord length, virtual wing mass, center of wing mass, and radial moment of inertia of the wing) were measured immediately after the insect was frozen. The methodology for measuring these morphological parameters is given in detail in [16].

### 2.4. Kinematics 

We selected flight sequences in which the butterfly flew steadily forward over the lake. In order to reconstruct the three-dimensional motion of the body and wings, we further selected flight sequences for which the insect remained in the field of view of both cameras for at least one wing stroke. The configuration of the cameras restricted the field of view to a rectangular volume of dimensions 20 cm × 20 cm × 16 cm (length, width, and height), i.e., approximately 4 wing lengths wide. Flight sequences (21–38 frames) from two female and two male *Heliconius sara* flying over the lake met these criteria.

In each synchronized camera field, we used QuickImage (a modification of NIH Image written by Jeff Walker, University of Southern Maine) to digitize the two-dimensional (x, y) position of the front of the head, the positions where each forewing intersected the thorax, the posterior tip of the abdomen, and the two forewing tips. For each frame the terrestrial horizon was digitized using the point on each side of the video frame where the lake met the shore. We corrected for error due to camera perspective using the position in one camera to determine the distance from the center of the second camera’s lens (and vice-versa). Data were imported into a Matlab program, and the digitized coordinates were then rotated about each frame horizon such that gravity was vertically downward, and the true horizon was orthogonal to this direction (Appendix A). Finally, coordinates were transformed so that the origin lay on the thorax between the wing bases. From the origin, three axes were defined: a vertical axis (*z*-axis), a horizontal axis perpendicular to the longitudinal body axis (*x*-axis), and a horizontal axis parallel to the longitudinal body axis (*y*-axis).

We measured the elevation of both wings and body angle relative to a horizontal plane passing through the thorax. We assumed the near and far wings were flapping symmetrically in steady forward flight, and calculated the roll of the insect as one-half the difference between the elevation of the far and near wing (the roll was positive when towards the near wing). Wing tip coordinates were then adjusted for the roll (rotating x and z about the *y*-axis). Taking these coordinates from sequential frames, we then calculated the stroke plane angle (SPA) using reduced major axis regression [17] of the wing tip elevation on its position anterior or posterior to the wing base (SPA = 90° − atan (slope of *z* on *y*)). The wing coordinates were then adjusted for SPA by rotating *y* and *z* about the *x* axis such that the *y*-axis was parallel to the SPA. Following rotation, wing elevation (Φ = atan [*y*/*x*]) was positive when dorsal to the wing base.

The kinematic data for wing elevation over time were smoothed using a Fourier analysis (Figure 1, for mathematical details, see [18]). From this, we calculated the mean elevation of the wing. Wing motion was characterized as the summation of the fundamental and first harmonic sine waves. The mean elevation and Fourier coefficients to describe the wing kinematics were included in the subsequent aerodynamic analysis.

### 2.5. Additional Kinematic Data

We also used data collected during 1988–1990 for one female and three male *H. sara* filmed flying through a hardware cloth tunnel in an insectary or greenhouse. These flight sequences (21–29 frames) were videotaped at 60 fps from a single camera. Methods for video recording and reconstruction of the wing position relative to the wing base from a single camera view are explained in detail in [19]. Analysis of wing kinematics was identical to that given above (Figure 2).

### 2.6. Aerodynamics

Based on the morphology and wing motion of each individual butterfly, we used a blade-element analysis based on quasi-steady aerodynamics to calculate the power required for the insect to fly forward [20]. In brief, aerodynamic power requirements for flight *P*_aero_ [21,22] may be divided into the power required to overcome drag on the wings (profile power: *P*_pro_), the power required to overcome drag on the body (parasite power: *P*_par_), and the power required to balance the body weight (induced power: *P*_ind_). *P*_pro_ was estimated as the product of profile drag and relative velocity for each element of the wings. In order to estimate the mean lift coefficient, we assumed that the vertical force produced by the wings balanced the body weight (i.e., vertical forces produced by the body were negligible) and no lift was produced during the upstroke.

The mean lift coefficient was calculated using the formula: mg = C_L_ (A_z,d_) + C_D,pro_ (B_z,d_ + B_z,u_)(1)
where m is body mass, g is acceleration due to gravity, C_L_* A_z,d_ equals the resolved lift force in the vertical direction over the downstroke, and C_D,pro_ *B_z,d_ and C_D,pro_*B_z,u_ equal the vertical components of the drag force over the downstroke and upstroke, respectively. We assumed the coefficient of drag C_D,pro_ was a function of the mean lift coefficient (C_D,pro_ = C_L_/1.73, [23,24,25]) for an active downstroke and it was a function of the Reynolds number (C_D,pro_ = 4.8/Re^0.5^) for an inactive upstroke [21,22]. The Reynolds number for the wings is Re = Uc/v, where U is the relative velocity of the mean chord c and v is the kinematic viscosity of air (1.55 × 10^−5^ m^2^s^−1^ at 23 C).

Induced power *P*_ind_ was estimated using momentum jet theory after [26]: *P*_ind_ = w_0_ (mg − B_z,d_ C_D,pro_ − B_z,u_ C_D,pro_).(2)

Induced velocity generated by the flapping wings was estimated as:w_0_ = [−V^2^/2 + (k^4^w^4^_RF_ + V^4^/4)^0.5^]^0.5^(3)
where V is the forward velocity (i.e., airspeed), k is a constant, and w_RF_ is the Rankin-Froude estimate of the induced velocity during hovering (for discussions of the constant k and w_RF_, see [21,22]).

The power required to overcome drag on the wings *P*_pro_ was estimated as the product of profile drag and the relative velocity for each element of the wing summed across the span and average over the course of the downstroke or upstroke.

Parasite power *P*_par_ was estimated by the product of the parasite drag and forward flight speed. We assumed parasite drag coefficient C_D,par_ = 0.2 [20]. Although airflow around the body is complicated by flapping flight, *P*_par_ is small relative to *P*_pro_ and *P*_ind_ at flight speeds measured in this study.

From these three estimates of power, the total aerodynamic power *P*_aero_ was calculated (Appendix A). Because of the high profile drag of the wings, *P*_aero_ is greater than the power required to accelerate the wings (*P*_acc_). Thus, *P*_aero_ during the decelerating halves of the downstroke and upstroke could not be supplied by the excess kinetic energy of the decelerating wing (*P*_acc_), and the total power requirement for flight is simply equal to *P*_aero_. The upstroke was assumed to be inactive which generally minimizes profile drag on the wings and thereby energy costs [20].

### 2.7. Prediction of Tailwind Drift Compensation for Passion-Vine Butterflies

In order to maximize migratory range, a butterfly should decrease its airspeed in a tailwind and increase it in a headwind. Quantitatively, this predicted airspeed is denoted as maximum range velocity, V_mr_. To quantify the relationship between V_mr_ and drift, we required an equation for the power curve where the velocity is greater than the velocity that requires the minimum power (minimum power velocity, V_mp_). For V_mr_ −V_mp_ > 0, we regressed *P*_aero_ on (V_mr_ − V_mp_)^k^ varying the exponent k by steps of 0.1 to find the equation *P*_aero_ = b_0_ + b_1_ (V_mr_ − V_mp_)^k^ that best explained the variance in *P*_aero_. From the best-fit curve, drift for each velocity of maximum range V_mr_ along the curve equaled: V_mr_ − *P*_aero_/[b_1_·k (V_mr_ − V_mp_) ^k−1^](4)

### 2.8. Measurement of Tailwind Drift Compensation

We measured wind drift and decomposed this vector into its crosswind and tailwind components for 18 *Heliconius sara* flying across Lake Gatún (Appendix A). In general, the method for measuring airspeed and track direction of *H. sara* and ambient wind speed and direction were the same as that given above (see *Filming of natural migratory flight*). Using vector analysis ([27], we calculated the groundspeed of each butterfly and decoupled the wind vector components perpendicular to the insect’s track (crosswind) and that parallel to the track (i.e., headwind or tailwind). For 15 butterflies, we also estimated their groundspeed by measuring the groundspeed of the boat with a marine GPS as we paced the insect’s horizontal flight speed. As a measure of error in the methods, we compared these data with those obtained by vector analysis (Figure 3).

## 3. Results

### 3.1. Wing Kinematics and Aerodynamic Power Relative to Velocity 

The regression of wing beat frequency (f) on airspeed (v) was not significant (n = 8, *p* = 0.759, Table 1). Stroke plane angle (SPA) tended to increase with airspeed (*p* = 0.074, SPA = 56.9° + 9.1 v, where v is in m/s), and the mean elevation of the wing (Φ) decreased significantly with airspeed (*p* = 0.0022, mean Φ = 20.2°−5.5 v). Stroke amplitude did not vary with either airspeed (*p* = 0.303) or wing beat frequency (*p* = 0.180). Stroke plane angle did not decrease with wing beat frequency (*p* = 0.181). The proportion of the wing stroke engaged in the downstroke did not vary significantly with wing beat frequency (*p* = 0.282) or with airspeed (*p* = 0.504). Wing tip paths of butterflies flying forward were consistent from one individual to the next (Figure 4). The mean lift coefficient (C_L_) decreased with airspeed (Figure 5A). A “J”-shaped curve best described the changes in aerodynamic power with forward velocity over the range of speeds measured (Figure 5B).

### 3.2. Predicted Relationship between Airspeed and Tailwind Drift Compensation 

The best-fit equation for the right-hand side of the J-shaped power curve (i.e., where airspeed is greater than V_mp_) was *P*_aero_ = 13.188 + 6.508 (V_mr_ − V_mp_) ^1.4^ (n = 4, R^2^ = 0.996, *p* < 0.005). As a result, in order to maximize its range, a butterfly should decrease V_mr_ as tailwind drift increases. V_mp_ serves as an asymptote (i.e., V_mr_ > 0.9), and when there is no drift, predicted V_mr_ = 2.25 m/s.

### 3.3. Do Passion-Vine Butterflies Compensate for Wind Drift?

Crosswind drift blows an insect off-course. Those butterflies that flew southwesterly (tracks between 164°–348°) drifted significantly downwind (n = 14, track = 245° − 1.05 drift in degrees, *p* = 0.0095, R^2^ = 0.442). Thus, there was no evidence for compensation for crosswind drift. Without compensation, butterflies are not expected to change airspeed with crosswind drift [28].

Headwind and tailwind drift alter the rate at which the insect flies across its migratory ground. Airspeeds of *Heliconius sara* declined with tailwind drift (n = 18, tailwind drift = 1.56 – 0.45 · airspeed). The 95% confidence intervals about the mean overlapped the predicted regression of tailwind drift on airspeed based on the power curve for *H. sara* (Figure 6). However, it should be noted that the regression was also not significantly different from a slope of zero, i.e., a lack of compensation for tailwind drift (*p* = 0.346).

The mean airspeed of male *H. sara* flying naturally across the Panama Canal was 2.9 m/s ± 1.0 (s.d., range 1.0 to 4.2 m/s, n = 10), and that for female *H. sara* was 2.7 m/s ± 1.0 (range 1.0 to 4.3 m/s, n = 15). On average, males and females were migrating at speeds 0.65 and 0.45 m/s faster than the predicted V_mr_ when there was no drift (2.25 m/s).

## 4. Discussion

These data combine a number of features of *Heliconius sara* for the first time. We have measured the kinematics of *H. sara* butterflies flying freely across the Panama Canal, and we characterized changes in wing kinematics over a range of natural flight speeds. As velocity of *H. sara* increased the head pitched down, increasing the stroke plane angle from more horizontal to more vertical, resulting in greater thrust to overcome drag on the wings and body. These results are in general agreement with those of the hawkmoth *Manduca sexta* [29]. Mean elevation of the butterfly’s wings decreased with velocity, whereas in *M. sexta* it showed a slight increase. At the top of the upstroke, the angle defined by the forewing tip to the thorax to the forewing tip was ca. 65° (Figure 1), too distant for lift by clap and fling characteristic of other butterflies [30,31].

At airspeeds greater than 1.75 m/s, the coefficients of lift were less than one and near to values measured for a variety of insect wings in steady airflow. Higher values of C_L_ are derived from unsteady flows associated with high lift mechanisms acting on flapping insect wings [24,25]. In addition, changes from an inactive upstroke, which minimizes drag at higher airspeeds, to an active one enhances vertical lift and weight support at lower velocities but at a cost of increased drag on the wings [30]. Here we assumed an inactive upstroke at all velocities, which given the geometry of the wing motion is a reasonable assumption.

In testing for adaptive wind drift in *H. sara*, we have assumed that the aerodynamic power curve is quantified accurately. Butterflies migrating freely over the lake were very difficult to videotape on both cameras simultaneously. The limited sample size also limits confidence in the interpolated curve. Yet a comparable, three-dimensional data set characterizing the kinematics of another aerial organism migrating naturally does not exist. In addition to the intraspecific variation in flight speed of *H. sara*, relationships between flight speed, morphology [16,32,33], and kinematics among butterfly species would benefit from the simultaneous measurement of flight speed and kinematics of insects flying naturally across the lake utilized here. Convergence in kinematics of distantly-related species in the context of locomotor mimicry has been measured from a single camera view of butterflies flying over the lake [12], but dual cameras yield more accurate kinematic reconstructions. At least one pierid butterfly had passive deformation of the wings (camber) that could play a role in generating its characteristic erratic flight pattern. With three-dimensional reconstruction of the leading and trailing edges of each wing pair, wing camber and attack angles of the wing chords integrated across the wingspan could be related to forward velocity. Several synchronized cameras in a broader configuration would increase the field of view where at least two orthogonal cameras intersect and permit the complete stroke cycles of more erratic butterfly flight patterns to be visualized in slow motion for analysis of wing kinematics. Despite the difficulties, further evaluation of free-flying, naturally migrating insects (e.g., drift-compensating *Pantala* dragonflies [6]) is needed to advance our understanding of insect flight. With improvement in the field of view, this methodology opens the full diversity of kinematics exhibited by a variety of naturally flying insects to exploration.

The blade element analysis is also very sensitive to assumptions of the coefficient of drag. For example, Srygley and Ellington [20] applied a C_D_ proportional to the Reynolds number to the active half-stroke in four other *Heliconius* species. When this assumption is applied to *H. sara*, the power required to fly was only one-third the magnitude of the values reported in Figure 4B. The curves were described by the same exponent (k = 1.4), and were thus similar in shape, but predicted V_mr_ was less (V_mr_ = 1.15 m/s) reflecting the lower position of the curve based on C_D_ proportional to the Reynolds number. *H. sara* in the field flew at velocities around 2.8 m/s, much closer to that predicted by the power curve based on a C_D_ proportional to the coefficient of lift. For this taxon, at least, we may be following the right track, but sensitivity of the power curve to C_D_ underscores the need for additional empirical studies to validate theory.

An extensive literature on the selection of optimal migratory airspeeds in birds has grown from theoretical predictions based on the U-shaped power curve of flight (see [34] for theoretical developments). However, the aerodynamic power curve for birds has rarely been quantified directly [35]. Direct measurement of the pectoralis muscle power suggests that the power curve may be U- or J-shaped or have a very flat mid-range (ca. flattened-U or square-shaped) depending on the species examined and the relative contribution of the power components as velocity changes [35]. Energetic expenditure by flight muscles other than the pectoralis major have not been assessed. Despite the lack of quantitative predictions, theoretical predictions of tailwind drift compensation are probably qualitatively correct [36,37,38].

The aerodynamic power curve of *H. sara* is the first to be quantified for butterflies. The only other power curves for Lepidoptera are those of *Urania fulgens*, a diurnal moth, and *Manduca sexta*, a hawkmoth. The aerodynamic power curve for *H. sara* is best described as J-shaped over the range of speeds tested with a minimum power requirement at airspeeds near to 0.9 m/s. Due to their larger masses, values of V_mp_ for *M. sexta* [29] and *U. fulgens* [9] were both greater than *H. sara*. Interestingly when the butterflies had an abundance of food and were flying locally within an insectary, they adopted the velocity near to the minimum power requirement (cf. the four points at or below minimum power velocity in Figure 5B). This would maximize the time spent flying and searching for receptive mates and hostplants between feeding bouts.

With a U- or J-shaped power curve, butterflies maximizing their range of migration should decrease their airspeed (V_mr_) in a tailwind and increase it in a headwind. Aerodynamic power increased with velocity (to the power of 1.4). Therefore, flying at airspeeds greater than predicted V_mr_ is inherently costly. The change in *H. sara*’s airspeeds with tailwind drift was not significantly different from that predicted by the J-shaped power curve. However, butterflies had sufficient variation in airspeeds that we were unable to exclude a lack of compensation for tailwind drift altogether. This null hypothesis would be consistent with the lack of compensation for crosswind drift evident as the butterflies migrated across the lake.

The tracks of migrating *H. sara* were, on average, towards 48° and 228°. It is evident that the majority were heading towards 245° and were blown off-course. In central Panama during May-July, we have observed a variety of pierid and nymphalid butterfly species and *Urania* moths migrating generally in a southwesterly direction [27]. The flyway crosses the isthmus of Panama at the Panama Canal, where the butterflies migrate from the wet Atlantic coastal forest to the drier Pacific forests annually after the onset of the rainy season. Ecological factors that drive these lepidopteran migrations include temporal and spatial variation in temperature, precipitation, soil moisture, and hostplant leaf-flushing driven by the El Niño Southern Oscillation [39,40]. Failure to compensate for crosswind drift may thus result in inadvertent displacement and potentially adverse consequences for survival and reproduction.

## Figures and Tables

**Figure 1 insects-14-00112-f001:**
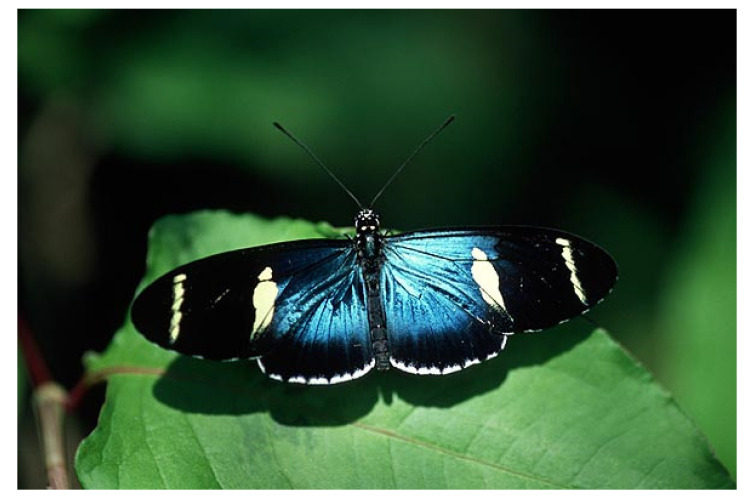
*Heliconius sara*.

**Figure 2 insects-14-00112-f002:**
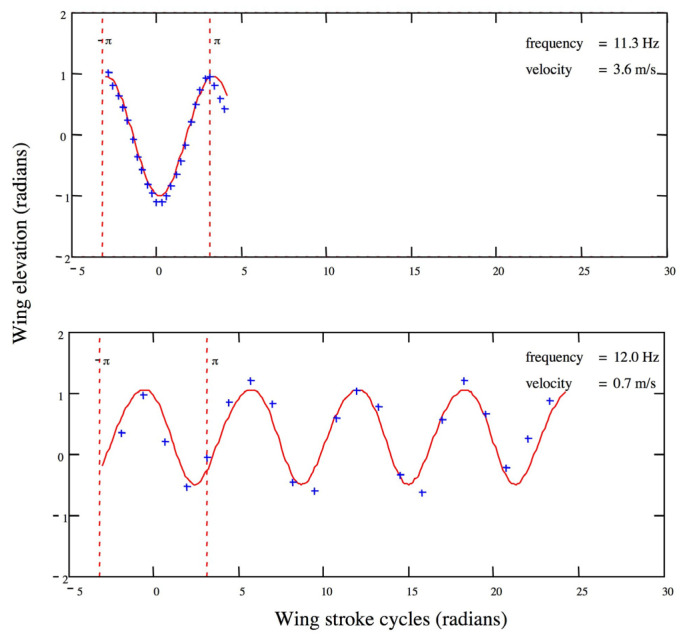
Fourier reconstruction of wing strokes. At the top, a single wing stroke of a *H. sara* filmed at 250 fps. Below, multiple wing strokes of a butterfly filmed at 60 fps. Wing elevation is 0 radians when horizontal. Blue crosses: raw data, red lines: best-fitting Fourier transform of the data.

**Figure 3 insects-14-00112-f003:**
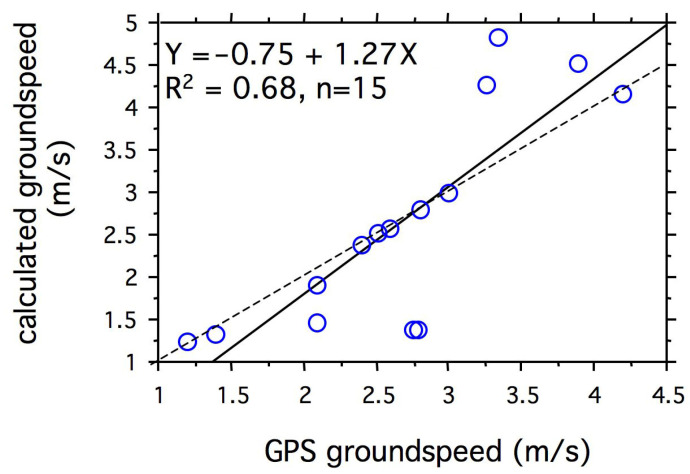
Groundspeed calculated from the vector analysis of airspeed, track over ground, wind speed, and wind direction relative to groundspeed measured with a Garmin marine differential GPS. The solid line shows the regression of the calculated values on those measured, and the dashed line indicates a one-to-one correspondence.

**Figure 4 insects-14-00112-f004:**
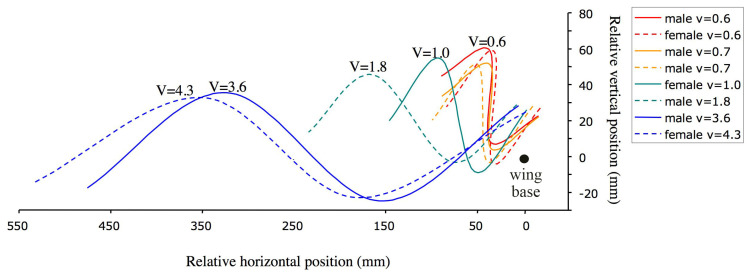
Wing tip paths of *H. sara* butterflies flying at forward velocities, V. Paths are drawn for 1.5 cycles from the beginning of the downstroke with the wing base positioned at the origin (solid circle). We assumed that the induced and horizontal velocities did not vary in the downstroke and upstroke. Stroke plane angle and the mean elevation of the wing declined significantly with airspeed. At approximately 4.0 m/s, the butterflies are traversing forward 10 wing lengths (350 mm) per cycle.

**Figure 5 insects-14-00112-f005:**
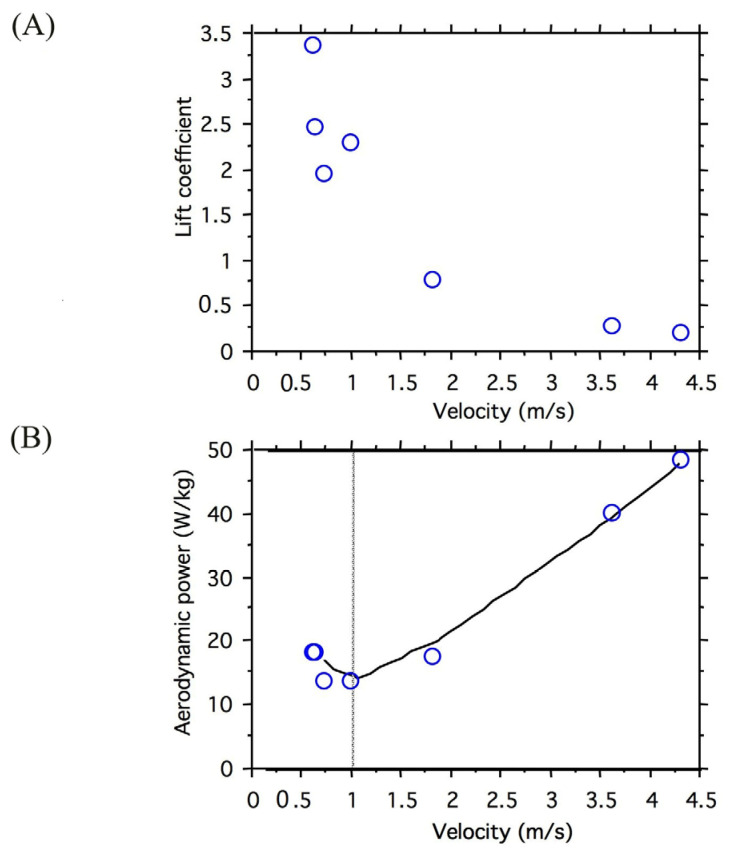
(**A**) Lift coefficient relative to forward velocity for *H. sara* butterflies; (**B**) Aerodynamic power relative to forward velocity for *Heliconius sara* butterflies. Each point represents a different butterfly. The dotted line is drawn at the minimum power velocity (V_mp_).

**Figure 6 insects-14-00112-f006:**
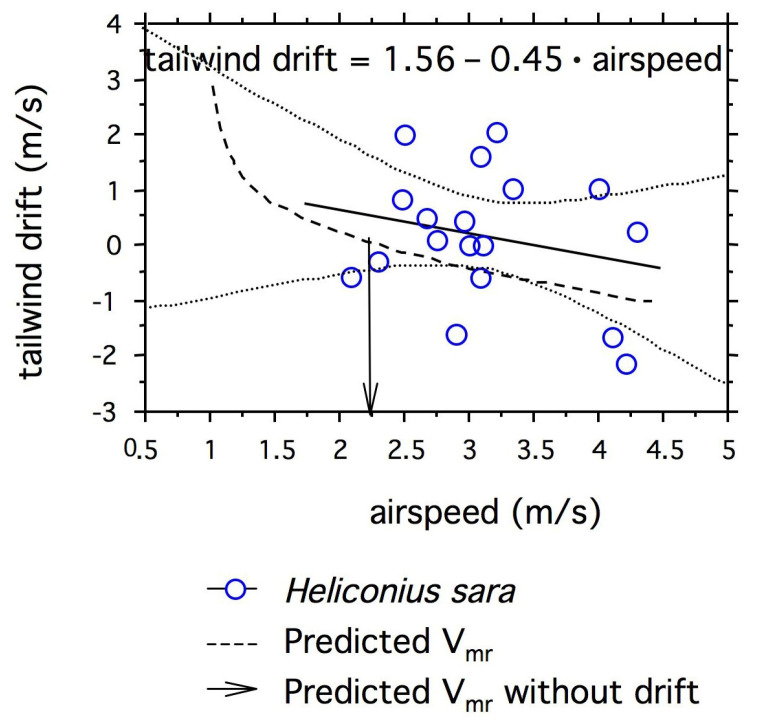
Tailwind drift relative to airspeed for *H. sara* migrating over Lake Gatún. Calculation of drift for values of V_mr_ > V_mp_ yielded tailwind drift with negative values and headwind drift with positive values. Because tailwinds assist the groundspeed, we have reversed the signs. The dashed line indicates the predicted curvilinear regression based on the aerodynamic power curve and a coefficient of drag proportional to the lift coefficient. The linear regression for drift of individual *H. sara* (open circles) versus their airspeeds is shown as a solid line and the 95% confidence limits about the means are shown as dotted lines.

**Table 1 insects-14-00112-t001:** Sex, velocity, mass and wing kinematics for *Heliconius sara* butterflies.

Sex	V (m/s)	Mass (mg)	Wing Beat Frequency (Hz)	Stroke Plane Angle (°)	Stroke Amplitude (°)	Mean Wing Elevation(°)
Female	4.3	154.1	12.2	80	95	−3
Male	3.6	117.6	11.3	97	112	−1
Male	1.8	94.4	11.5	101	83	14
Female	1.0	101.1	10.5	78	92	7
Male	0.7	89.3	13.5	50	80	17
Male	0.7	68.8	12.0	66	89	19
Female	0.6	117.4	12.6	53	107	14
Male	0.6	82.0	11.3	51	88	21

## Data Availability

Data are in the Appendix A.

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
