# Peer review of "Quantifying the Aerodynamic Power Required for Flight and Testing for Adaptive Wind Drift in Passion-Vine Butterflies Heliconius sara (Lepidoptera: Nymphalidae)"

_insects, 2023, doi:10.3390/insects14020112_

Round 1

Reviewer 1 Report

The manuscript estimates aerodynamic power from video of free flying butterflies in the field and tests for an optimal/adaptive migration strategy of accounting for head/tail wind drift. 

While I admire the effort to use data collected in the field (natural migration flight) I don't think the data collected and their accuracy are sufficient to inform on the aerodynamic power and its relationship with flight speed. Since the section on tail wind drift is based on the power analysis its conclusion are questionable as well. 

For the flapping kinematics the sample size is low (4 butterflies) and has high uncertainty (filming from a moving boat) and unclear accuracy (error in measuring boat and air speed). Another set of data for the kinematics was obtained at low filming speed (60 Hz) and using a single camera. In both cases there is no measurement of the orientation of the wing chord relative to the flight or flapping speed and how it changes with the air speed. Then, the data are used to estimate aerodynamic power with numerous assumptions that are justified by studies on other insects but not on the current one. This kind of data on flapping kinematics can be used in behavioral analyses or maybe even to conjecture on flight energetics but the sample size and accuracy do not permit to reasonably evaluate the relationship between flight speed and power. The reported relationship between the two is simply a result of the assumptions in the estimation method (dependence on Re number and assuming all other parameters remain unchanged with flight/air speed.) 

To name just a few difficulties:

 - The movies from the boat were corrected for pitch (horizon, line 138) how were they corrected for heave and roll (of the boat)?

-what were the flight speeds in the 4 movies used for the kinematics and how are they compared to the flight speeds in table S2 ?

- How was the boat speed, butterfly speed and wind velocity synchronized (they all vary in time)?

-Why isn't the pitch of the body (line 307) included in the parasite power calculation (line 206)?

- The assumption of CL = 1.73 CD_pro is not based on data for this species and does not take into account the interaction of the wing with the forward flight speed. Yet it is used to estimate differences in power with flight speed. Lines 329-336 demonstrate nicely why this can be a big problem arguing against making the conjectures (about the shape of the power curve) made in this manuscript  

In summary, although the authors made a genuine effort, the small sample size, uncertain accuracy and missing parameters forcing sensitive (and sometimes unbased) assumptions combine to make the conclusions made here questionable. The conclusions are simply not supported by sound data.          

Author Response

While I admire the effort to use data collected in the field (natural migration flight) I don't think the data collected and their accuracy are sufficient to inform on the aerodynamic power and its relationship with flight speed. Since the section on tail wind drift is based on the power analysis its conclusion are questionable as well. 

For the flapping kinematics the sample size is low (4 butterflies) and has high uncertainty (filming from a moving boat) and unclear accuracy (error in measuring boat and air speed). Another set of data for the kinematics was obtained at low filming speed (60 Hz) and using a single camera. In both cases there is no measurement of the orientation of the wing chord relative to the flight or flapping speed and how it changes with the air speed. Then, the data are used to estimate aerodynamic power with numerous assumptions that are justified by studies on other insects but not on the current one. This kind of data on flapping kinematics can be used in behavioral analyses or maybe even to conjecture on flight energetics but the sample size and accuracy do not permit to reasonably evaluate the relationship between flight speed and power. The reported relationship between the two is simply a result of the assumptions in the estimation method (dependence on Re number and assuming all other parameters remain unchanged with flight/air speed.) 

Reply: Measurements for the orientation of the wing chord (i.e., the attack angle) would be an excellent addition to this study. There are suitable landmarks on the forward and leading edges of many butterfly wings that might be followed to calculate attack angle. Some species other than Heliconius sara showed passive deformation of the wings (camber) that would be fascinating to investigate with reconstruction of the entire leading and trailing edges of the wing pairs. We would hope to be able to do this in future studies, but for this paper, we do not have information on attack angle. To draw attention to these possibilities, I added the following sentences to the Discussion: ‘At least one pierid butterfly had passive deformation of the wings (camber) that could play a role in generating its characteristic erratic flight pattern. With three-dimensional reconstruction of the leading and trailing edges of each wing pair, wing camber and attack angles of the wing chords integrated across the wing span could be related to forward velocity.’

To name just a few difficulties:

 - The movies from the boat were corrected for pitch (horizon, line 138) how were they corrected for heave and roll (of the boat)?

Reply: Each frame is corrected for pitch in the horizon, and since the cameras are at 90 degrees to one another, this corrects both images for the ‘heave and roll’ of the boat and the person in the boat holding the steel bar with the cameras mounted on it.

-what were the flight speeds in the 4 movies used for the kinematics and how are they compared to the flight speeds in table S2 ?

Reply: The four fastest velocities in Table 1 correspond to the butterflies with video images from two synchronized cameras. Only the female with an airspeed of 4.3 m/s, is included in Table S2. A second female was filmed following release on the lake and thus would not be expected to compensate for wind drift. Directional data were not collected for the two males at 1.8 and 3.6 m/s, and so drift could not be calculated.

- How was the boat speed, butterfly speed and wind velocity synchronized (they all vary in time)?

Reply: Airspeed was collected with the unidirectional anemometer at the same time as the butterfly was filmed. So those two measurements are synchronized. Wind velocity and direction were collected immediately following capture of the butterfly, as stated in the text.

-Why isn't the pitch of the body (line 307) included in the parasite power calculation (line 206)?

Reply: Fortunately, parasite power is a very minor component of the total aerodynamic power required to fly (less than 1%). As explained in the text, the vortex wake makes estimating the circulation around the body inaccurate. As a result, we use the simplified calculation with the stated caveats.

- The assumption of CL = 1.73 CD_pro is not based on data for this species and does not take into account the interaction of the wing with the forward flight speed. Yet it is used to estimate differences in power with flight speed. Lines 329-336 demonstrate nicely why this can be a big problem arguing against making the conjectures (about the shape of the power curve) made in this manuscript  

Reply: The coefficient of drag varies with forward flight speed, and so CL does take into account interaction of the wing with forward flight speed.

In summary, although the authors made a genuine effort, the small sample size, uncertain accuracy and missing parameters forcing sensitive (and sometimes unbased) assumptions combine to make the conclusions made here questionable. The conclusions are simply not supported by sound data.      

Reply: We understand that the sample size is small and we have stated in the Discussion that the limited sample size limits confidence in the interpolated curve. Yet a comparable three-dimensional data set characterizing the kinematics of another aerial organism migrating naturally does not exist. Therefore we agree that additional research is needed to fortify the conclusions. Following the suggestion of another reviewer, we have outlined in the Discussion how the methods might be applied to investigate kinematic diversity exhibited by naturally flying insects more generally.

Reviewer 2 Report

The article is a significant and novel contribution to the understanding and study of flight in butterflies and insects in general. Unfortunately, articles that bring us closer to unraveling adaptive flight behaviors in butterflies (and insects) under natural conditions and their association with flight kinematics and wing morphology are scarce.  So it was a pleasure to read it, and I have a few comments.

The authors must use some space in the introduction to contextualize the limited data in the literature on the flight in butterflies under natural conditions (e. g. Dudley & Srygley, 1994) and how important it is to have this information and its relationship to flight kinematics. Even in terms of citations and the overall impact of the article, it would be important to highlight some methodological suggestions by the authors in the discussion section that would help to increase this type of data on butterflies (and insects in general) in the near future.

References are missing for lines 68-71

Table 1 appears before being mentioned in the text.

Lines 247-248. The variable names must coincide with those shown in table 1.

Figure 4. Move the black circle that appears over "wing base." 

In Table S1, correct the date column. 

In Table S2 some column headings could be more intuitive.

Author Response

The article is a significant and novel contribution to the understanding and study of flight in butterflies and insects in general. Unfortunately, articles that bring us closer to unraveling adaptive flight behaviors in butterflies (and insects) under natural conditions and their association with flight kinematics and wing morphology are scarce.  So it was a pleasure to read it, and I have a few comments.

The authors must use some space in the introduction to contextualize the limited data in the literature on the flight in butterflies under natural conditions (e. g. Dudley & Srygley, 1994) and how important it is to have this information and its relationship to flight kinematics. Even in terms of citations and the overall impact of the article, it would be important to highlight some methodological suggestions by the authors in the discussion section that would help to increase this type of data on butterflies (and insects in general) in the near future.

Reply: We have already stated that ‘quantitative analysis of changes in kinematics and aerodynamics with flight speed are lacking for any butterfly species in free, natural flight’ in the Introduction, citing more recent papers than Dudley and Srygley 1994. We prefer to cite Dudley and Srygley 1994 in the Discussion where we have followed the Reviewer’s suggestion and highlighted several observations that might motivate researchers to obtain more kinematic data on free-flying butterflies and other migrating insects as well as constraints that we encountered. We have added the following text and citations to the Discussion: ‘In addition to the intraspecific variation in flight speed of H. sara, relationships between flight speed, morphology [16, 32-33], and kinematics among butterfly species would benefit from the simultaneous measurement of flight speed and kinematics of insects flying naturally across the lake utilized here. Convergence in kinematics of distantly-related species in the context of locomotor mimicry has been measured from a single camera view of butterflies flying over the lake [12], but dual cameras yield more accurate kinematic reconstructions. At least one pierid butterfly had passive deformation of the wings (camber) that could play a role in generating its characteristic erratic flight pattern. With three-dimensional reconstruction of the leading and trailing edges of each wing pair, wing camber and attack angles of the wing chords integrated across the wing span could be related to forward velocity. Several synchronized cameras in a broader configuration would increase the field of view where at least two orthogonal cameras intersect and permit the complete stroke cycles of more erratic butterfly flight patterns to be visualized in slow motion for analysis of wing kinematics. Despite the difficulties, further evaluation of free-flying, naturally migrating insects (e.g., drift-compensating Pantala dragonflies [6]) is needed to advance our understanding of insect flight. With improvement in the field of view, this methodology opens the full diversity of kinematics exhibited by a variety of naturally flying insects to exploration.’ Two citations have been added to the text: [32] Dudley and Srygley 1994 and [33] LeRoy et al. 2019.

References are missing for lines 68-71

Reply: This statement from line 68=71 is also from citation [3]. To clarify, we cited [3] again after line 71.

Table 1 appears before being mentioned in the text.

Reply: We ask the Editor if this format is a problem for the journal. We prefer to have the Table on the same printed page as the Results text which refers to it.

Lines 247-248. The variable names must coincide with those shown in table 1.

Reply: Corrected.

Figure 4. Move the black circle that appears over "wing base." 

Reply: The black circle is at the correct coordinates: 0, 0.

In Table S1, correct the date column.

Reply:Done.

In Table S2 some column headings could be more intuitive.

Reply: Fixed.

Reviewer 3 Report

The manuscript presents measurements of flight trajectories and wing kinematics of migrating passion-vine butterflies. Based on the measured data, aerodynamic power and maximum range velocities are calculated, in presence of tailwind drift. The manuscript is well written and contains valuable data. I think it can be published after a minor revision.

Specific comments.

1.       Figure 5 in page 9 should be renamed into figure 6.

2.       The horizontal axis in figure 5 page 8 is labelled ‘velocity’, in figure 5 page 9 it is ‘airspeed’. Shouldn’t it be the same quantity? If so, why use two different names?

3.       Trough the text: Reynold’s number -> Reynolds number or Reynolds’ number.

4.       Page 362: “increased exponentially with the velocity (to the power 1.4)” -> increased algebraically with the velocity (with the exponent 1.4)”. Exponentially means P ~ exp(V).

5.       Simple Summary: the word “exponentially” (used twice) should be replaced with a technically current word but not a mathematical term. Rapidly?

6.       Page 9, line 275: again, the word “exponentially” is used incorrectly.

7.       Page 6, line 218: ”In order to maximize migratory range, a butterfly should decrease its airspeed in a tailwind and increase it in a headwind.” The authors should explain why this should happen. I don’t find it obvious.

Author Response

The manuscript presents measurements of flight trajectories and wing kinematics of migrating passion-vine butterflies. Based on the measured data, aerodynamic power and maximum range velocities are calculated, in presence of tailwind drift. The manuscript is well written and contains valuable data. I think it can be published after a minor revision.

Specific comments.

  1. Figure 5 in page 9 should be renamed into figure 6.

Reply: Done.

  1. The horizontal axis in figure 5 page 8 is labelled ‘velocity’, in figure 5 page 9 it is ‘airspeed’. Shouldn’t it be the same quantity? If so, why use two different names?

Reply: This is a very good point. Aerodynamic power is based on forward velocity through the air, which is the airspeed of the insect, as opposed to its groundspeed. Drift is calculated with a vector analysis of airspeed and heading, ground speed and track, and wind speed and wind direction. So we use airspeed and velocity interchangeably in this paper. In line 199, we added (i.e., airspeed) after forward velocity to indicate that airspeed is the measure for forward velocity of the insects filmed over the lake for the analysis of their aerodynamic power requirements.

  1. Trough the text: Reynold’s number -> Reynolds number or Reynolds’ number.

Reply: Done.

  1. Page 362: “increased exponentially with the velocity (to the power 1.4)” -> increased algebraically with the velocity (with the exponent 1.4)”. Exponentially means P ~ exp(V).

Reply: Thank you to the reviewer for indicating this error. We deleted exponentially and changed it to ‘increased with velocity (to the power of 1.4)’.

  1. Simple Summary: the word “exponentially” (used twice) should be replaced with a technically current word but not a mathematical term. Rapidly?

Reply: We substituted ’rapidly’ for the first mention and ‘non-linearly’ for the second.

  1. Page 9, line 275: again, the word “exponentially” is used incorrectly.

Reply: We deleted exponentially from this sentence.

  1. Page 6, line 218: ”In order to maximize migratory range, a butterfly should decrease its airspeed in a tailwind and increase it in a headwind.” The authors should explain why this should happen. I don’t find it obvious.

Reply: This is based on maximizing range on a given amount of fuel as stated in the introduction. In a tailwind, the insect will remain aloft longer and thus achieve the greatest distance by reducing its airspeed (and the power required to fly), whereas in a headwind, the insect will achieve the greatest distance by increasing its airspeed and the power required to fly. Note this principle of adjusting airspeed to maximize range on a given amount of fuel applies to airplanes, as well. 

Round 2

Reviewer 1 Report

Thank you for the detailed reply but the same technical concerns I had about the previous version (accuracy, sample size) remain in this revised version. I'm afraid that these are problems that can't be solved by revising the text.